# Recovery-Stress Response of Blood-Based Biomarkers

**DOI:** 10.3390/ijerph18115776

**Published:** 2021-05-27

**Authors:** Sebastian Hacker, Thomas Reichel, Anne Hecksteden, Christopher Weyh, Kristina Gebhardt, Mark Pfeiffer, Alexander Ferrauti, Michael Kellmann, Tim Meyer, Karsten Krüger

**Affiliations:** 1Department of Exercise Physiology and Sports Therapy, Institute of Sport Science, Justus-Liebig-University Gießen, 35394 Gießen, Germany; sebastian.hacker@sport.uni-giessen.de (S.H.); Thomas.Reichel@sport.uni-giessen.de (T.R.); Christopher.Weyh@sport.uni-giessen.de (C.W.); kristina.gebhardt@sport.uni-giessen.de (K.G.); 2Institute of Sports and Preventive Medicine, Saarland University, 66123 Saarbrücken, Germany; a.hecksteden@mx.uni-saarland.de (A.H.); tim.meyer@mx.uni-saarland.de (T.M.); 3Institute of Sport Science, Johannes Gutenberg University Mainz, 55128 Mainz, Germany; Mark.Pfeiffer@uni-mainz.de; 4Department of Training & Exercise Science, Faculty of Sport Science, Ruhr University Bochum, 44801 Bochum, Germany; alexander.ferrauti@ruhr-uni-bochum.de; 5Unit of Sport Psychology, Faculty of Sport Science, Ruhr University Bochum, 44801 Bochum, Germany; michael.kellmann@rub.de; 6School of Human Movement and Nutrition Sciences, The University of Queensland, St. Lucia 4072, Australia

**Keywords:** cytokines, muscle damage, chemokines, badminton, soccer, monitoring

## Abstract

The purpose of this study was to investigate blood-based biomarkers and their regulation with regard to different recovery-stress states. A total of 35 male elite athletes (13 badminton, 22 soccer players) were recruited, and two venous blood samples were taken: one in a ‘recovered’ state (REC) after a minimum of one-day rest from exercise and another one in a ‘non-recovered’ state (NOR) after a habitual loading microcycle. Overall, 23 blood-based biomarkers of different physiologic domains, which address inflammation, muscle damage, and tissue repair, were analyzed by Luminex assays. Across all athletes, only creatine kinase (CK), interleukin (IL-) 6, and IL-17A showed higher concentrations at NOR compared to REC time points. In badminton players, higher levels of CK and IL-17A at NOR were found. In contrast, a higher value for S100 calcium-binding protein A8 (S100A8) at REC was found in badminton players. Similar differences were found for BDNF in soccer players. Soccer players also showed increased levels of CK, and IL-6 at NOR compared to REC state. Several molecular markers were shown to be responsive to differing recovery-stress states, but their suitability as biomarkers in training must be further validated.

## 1. Introduction

For many years the relationship between exercise response and recovery, as well as the concomitant impact on performance has attracted the interest of sport science [1]. Since elite athletes’ training loads and competition demands are increasing, an elaborate plus feasible exercise and recovery management, and a development of qualitative training parameters, is necessary to ensure high levels of performance [2]. Thus, it is of utmost importance that appropriate exercise stimuli are prescribed and (short- and long-term) physiological adaptation is achieved [3,4]. Improperly managed training loads may lead to accumulated fatigue and result in illness, injury, non-functional overreaching, and the overtraining syndrome [3,5]. Nonetheless, intensified training with subsequent exercise-induced fatigue is necessary to stimulate adaptation and enhance performance, indicating the necessity of systematic load and regeneration monitoring [3,6]. 

Exercise response, as well as recovery, are multi-faceted phenomena, which can be evident in several physiologic domains, such as hormonal, immunological, neuromuscular, or psychological [1,7]. For the purpose of our work, we chose blood-based biomarker regulations in the recovery-stress cycle and discipline-specific training performance as the most pertinent and objective exercise stimulus for elite athletes. However, as earlier studies stated, testing training or competition performance by the use of maximal strength tests can be seen as non-feasible due to the additionally induced fatigue [1]. Consequently, a wide variety of surrogate exercise response and recovery indicators have been investigated to detect the internal training load [8,9,10]. In particular, blood-based biomarkers may be favorable because of their high objectivity, precise measurement, reproducibility, and minimal training interference [11,12,13]. A problem with the use of biomarker analyses in the context of competitive sport is the lack of reference data and interindividual differences between athletes [14]. For this reason, currently available biomarkers are only partially suitable for training and regeneration management. This study represents a first step towards finding sensitive recovery and stress markers and drawing initial conclusions with regard to two different sport disciplines. 

There is insufficient evidence to demonstrate to what extent a single biomarker can reliably quantify the exercise response and recovery [8,15]. It is assumed that a panel of selected biomarkers may allow a more comprehensive, and sport specific analysis of athletic performance and recovery status [15,16]. Hence, regarding a thorough and serious assessment of athletes’ internal load state, the evaluation of physiological markers, which have already been shown in studies to be load-sensitive, seems promising. Thus, we compiled a panel of blood-based biomarkers reflecting several physiologic domains for further analysis. Because intensive exercise training represents a pro-inflammatory stimulus [17], the regulation of various interleukins and other pro- and anti-inflammatory proteins were analyzed. The specific selection of blood markers included here was based on preliminary data, in which some cytokines could be identified as exercise sensitive and reliable [15]. To gain insight into exercise-induced muscle damage we included enzymes like creatine kinase (CK), lactate dehydrogenase (LDH), myeloperoxidase (MPO), and various chemokines, which have already shown a certain sensitivity under athletic stress in previous studies [18]. Moreover, to measure potential variations in growth factors, we chose markers such as human growth hormone (GH), brain-derived neurotrophic factor (BDNF), or glial cell line-derived neurotrophic factor (GDNF) [7].

Thus, this study aimed to analyze the regulation of several physiologic domains including various chemokines, inflammatory cytokines, enzymes, hormones, proteinases and growth factors in athlete’s recovery-stress training cycles, as well as in clarifying their regulation within groups of badminton and soccer players.

## 2. Materials and Methods

### 2.1. Study Design and Participants

In total, 13 male badminton players, who prepared for the world championships and 22 semi-professional male soccer players, who played in the fourth German division (‘Regionalliga’) were recruited. Subjects of the badminton group were the same ones as included into the studies by Barth et al. and Schneider et al. [12,19]. Participants’ characteristics are presented in Table 1. Of each player two blood samples, one representing a REC state after a minimum of one-day rest from exercise and one representing a NOR state after a sport-dependent habitual loading microcycle. Thus, the athletes trained freely according to their current training schedules. The research team only joined the athletes at defined stages of training to collect the blood samples. For badminton players, this microcycle comprised four consecutive days with up to two sessions per day, including badminton specific, as well as endurance, strength and speed training. Daily training load consisted of 50- to 310-min of moderate to intensive exercise. Exemplary training plans can be found in a previously published study [19]. In case of the soccer players, training microcycle consisted of high-intensity sessions with strenuous small-sided games which could be quantified using a completed recovery-stress questionnaire [20]. At least two training sessions were completed within 48 h. Based on these monitoring controls, the training was characterized as an intensive soccer-specific training. With the exception of one participant, who was excluded from the analyses of IL-6 and S100A8 due to severe outlier data (Z score; IL-6: 5.75, S100A8: 5.74)*,* all participants were included in the final analysis. All participants were informed about the experimental procedures and provided their written informed consent. The study was approved by the local Human Research Ethics Committee (Ärztekammer des Saarlandes, approval no. 228/13 and amendments) and conducted in accordance with the Declaration of Helsinki.

### 2.2. Outcome Measures

To be considered for the panel, biomarkers had to fulfill certain criteria: markers (1) are proteins, which are measurable in blood plasma and serum, (2) are detectable through sensitive measuring techniques, preferably enzyme-linked immunosorbent assay (ELISA), and (3) are potentially exercise sensitive. Included blood-based biomarkers comprise renowned parameters like CK [8,21], as well as less frequent investigated or newly suggested markers for the recovery-stress response, e.g., S100A8 and S100A12 [22]. A summary of all physiologic categories and associated biomarkers analyzed in this study is given in Table 2. Those markers were chosen to reflect exercise sensitive processes related to inflammation, muscle damage, tissue repair and growth, and matrix remodeling.

### 2.3. Blood Sampling and Analyses

Venous blood samples were collected and consisted of 2.7-mL, as well as 2 × 9-mL anticoagulated EDTA plasma vacutainers. Centrifugation was conducted within 20 min after sampling. For blood sampling and concomitant analysis standard methods were used in accordance with previously published studies [11,12]. In detail, CK was analyzed by automated routine techniques (UniCell DxC 600 Synchron, Beckman Coulter GmbH, Krefeld, Germany). Blood sampling was scheduled on Monday and Friday between 7:45 and 8:30 A.M. As mentioned above athletes had a minimum of one day of rest before the REC sample was taken. Regarding NOR blood samples athletes had their last session before blood sampling in the afternoon/evening of Thursday. Thus, the range of duration between the last session and NOR blood sampling varied between 12 and 16 h. BDNF, CCL2, CCL4, CD163, GDNF, GH, IFN-γ, IL-10, IL-12p40, IL-17A, IL-1ß, IL-1RA, IL-6, IL-8, LDH, MMP-2, MMP-3, MMP-9, MPO, S100A8, S100A12, and SHBG were analyzed according to manufacturer’s instructions via a commercially available human magnetic Luminex assay (Bio-Techne Ltd., Abingdon, Oxon, UK) using the Luminex MAGPIX system (Luminex Corporation, Austin, TX, USA).

### 2.4. Statistical Analyses

Statistical analyses were carried out using JASP (Version 0.14.1, JASP Team, Amsterdam, The Netherlands). Normal or log-normal distribution of the mean differences was tested via the Kolmogorov-Smirnov test. For explorative analysis of marker regulation in all athletes at both status time points, as well as within sport disciplines Student’s paired *t* test and, in case of non-normal distributed data, the Wilcoxon signed-rank test were used. The level of significance was set at α = 0.05. Due to the explorative nature of our investigation, no adjustment of the alpha error for multiple testing has been carried out. Descriptive data are presented as raw data with mean ± SD. Additionally, mean differences or the Hodges-Lehman estimate, depending on the underlying test, as well as 95% confidence intervals (CI) are given. Tukey boxplots with raw data were created using Prism 9 (GraphPad Software, San Diego, CA, USA).

## 3. Results

### 3.1. Analyses of Blood-Based Biomarkers across Disciplines

CK showed higher enzyme activity at ‘non-recovered’ (NOR) time points compared to ‘recovered’ (REC) time points (*p* < 0.001; REC 152.6 ± 102.4 U/L; NOR 693.1 ± 435.5 U/L; mean difference 540.6 U/L; 95% CI [399.2, 681.9]) (Figure 1a). Similarly, IL-6 levels were higher at NOR state (*p* = 0.024; REC 12.5 ± 3.8 pg/mL; NOR 13.7 ± 3.7 pg/mL; mean difference 1.3 pg/mL; 95% CI [0.2, 2.4]) (Figure 1b). Concentrations of IL-17A were elevated at NOR compared to REC (*p* = 0.033; REC 38.8 ± 17.7 pg/mL; NOR 42.8 ± 19.6 pg/mL; Hodges-Lehman Estimate (HLE) 4.7 pg/mL; 95% CI [−1.2, 9.2]) (Figure 1c). CD163 increased at NOR; however, this marker did not reach statistical significance (*p* = 0.069; REC 1.0 × 10^6^ ± 650,929.7 pg/mL; NOR 1.2 × 10^6^ ± 794,388.1 pg/mL; mean difference 174,283.6 pg/mL; 95% CI [11,188, 337,379.3]) (Figure 1d). No differences were found for LDH, MPO, IFN-γ, IL-1ß, IL-1RA, IL-8, IL-10, IL-12p40, S100A8, S100A12, CCL2, CCL4, BDNF, GDNF, GH, SHBG, MMP-2, MMP-3, MMP-9 (Table 3).

### 3.2. Analyses of Blood-Based Biomarkers within Disciplines

#### 3.2.1. Cytokines

Badminton players had higher IL-17A concentrations at NOR compared to REC (*p* = 0.018; REC 38.7 ± 11.4 pg/mL; NOR 48.5 ± 25.2 pg/mL; HLE 6.6 pg/mL; 95% CI [1.9, 30.1]) (Figure 2b). In contrast, IL-6 levels of soccer players were higher at NOR compared to REC (*p* = 0.017; REC 12 ± 2.5 pg/mL; NOR 14 ± 2.9 pg/mL; mean difference 1.9 pg/mL; 95% CI [0.4, 3.5]) (Figure 3b). A similar trend between the time points was shown for IL-1ß concentration (*p* = 0.060; REC 54.7 ± 13.4 pg/mL; NOR 66.5 ± 27.6 pg/mL; mean difference 11.9 pg/mL; 95% CI [−0.5, 24.2]) (Figure 3c). No significant differences were found for IFN-γ, IL-1RA, IL-8, IL-10, and IL-12p40 (data not shown).

#### 3.2.2. Enzymes

For CK, significant differences between REC and NOR were found for badminton (*p* < 0.001; REC 117.4 ± 26.4 U/L; NOR 776.9 ± 585.4 U/L; HLE 495.5 U/L; 95% CI [359.5, 1131.5]) (Figure 2a), as well as soccer players (*p* < 0.001; REC 173.4 ± 124 U/L; NOR 643.6 ± 323.1 U/L; mean difference 470.3 U/L; 95% CI [351, 589.5]) (Figure 3a). No significant differences between both time points were found for MPO and LDH (data not shown).

#### 3.2.3. Chemokines, Growth Factors, Hormones, Other Inflammatory Signaling Molecules, and Proteinases

For S100A8 lower levels were found in the NOR state compared to REC state in badminton players (*p* = 0.047; REC 237.8 ± 52.5 pg/mL; NOR 206.4 ± 38.8 pg/mL; mean difference −23.8 pg/mL; 95% CI [−47.3, −0.3]) (Figure 2c). BDNF levels of soccer players showed a similar decrease in concentration at NOR compared to REC time-points (*p* = 0.043; REC 33,907.4 ± 5795.7 pg/mL; NOR 32,199.5 ± 6058.8 pg/mL; mean difference −1707.9 pg/mL; 95% CI [−3354.1, −61.7]) (Figure 3d). No significant differences were found for all other addressed chemokines, growth factors, hormones, proteinases, and further inflammatory signaling molecules (data not shown).

## 4. Discussion

The present study investigated the expression of numerous molecular markers in the blood in a specific exercise recovery cycle across sport disciplines and within sport disciplines. This distinguishes the work from other researchers, which mostly analyzed short-term responses after acute bouts of exercise [23,24,25]. It is relevant to consider this aspect when interpreting the data. This may be the reason why no differences were found for many markers that have been shown to be sensitive to stress in other settings. We selected the sports according to the fact that they are game sports, one of which contains more endurance aspects (soccer), the other contains shorter fast-paced actions. 

Our results suggest that CK, IL-6, and IL-17A can differentiate between REC and NOR time points in athletic populations. Moreover, a statistical trend was found for CD163 (*p* = 0.069). Within disciplines, analysis revealed additional parameters displaying sport specific biomarker alterations. In detail, badminton players showed significantly decreased levels of S100A8 after a habitual loading microcycle, whereas soccer players showed decreased BDNF levels. Another trend for IL-1ß (*p* = 0.060) was found, which may enable the characterization of REC or NOR states in soccer players, but this needs to be confirmed in future studies. However, a comparison between the sports is only possible to a limited extent because, within a microcycle, training is carried out using many different methods. This content was not subject to any specification and controlled in this study which is later discussed. Therefore, we cannot define the exact training content, but as an advantage, we are closer to the training reality.

Cytokine responses to exercise stimuli are known to be induced by muscle damage, but also independently of this through acute intense exercise loads, which instigate an inflammatory immune response and subsequently induce an elevation of various cytokines [26]. Many of these cytokines only increase for a short time and are then very quickly degraded [8,15]. This may be the reason why we found few significantly regulated markers in this setting. Only IL-17A and IL-6 unveiled significant differences between both conditions. This is, nevertheless, interesting because studies analyzing the exercise response of IL-17A in healthy or athletic populations are scarce. Our data prove a certain exercise sensitivity and, therefore, a potential suitability in athletic monitoring [27,28,29]. IL-17A is produced by a subset of CD4+ T cells, Th17 cells, which are part of the adaptive immune system and take a critical part in defense against extracellular bacterial and fungal infections [30]. Further, IL-17A induces various pro-inflammatory mediators, such as IL-1ß and IL-6, and might reflect muscle damage [27,30]. There is only little firm evidence on the effects of exercise on the production of IL-17. Sugama et al. [27] recruited 14 male triathletes, who participated in a duathlon race and analyzed urinary and plasma levels of IL-17. Urinary and plasma IL-17 concentrations decreased significantly immediately after exercise cessation. However, following a 3 h recovery phase, plasma IL-17 showed a significant increase in concentration compared to immediately post-exercise [27], which possibly indicates a delayed immunological response [28]. Kostrzewa-Nowak and Nowak saw a significant decrease in plasma IL-17A after participants had completed a 20 m shuttle run test, but not after completion of the Yo-Yo intermittent recovery test level 1 [29].

There are several studies which proved a certain exercise sensitivity of IL-6 [15,26,31]. IL-6 has pleiotropic characteristics and can induce pro- as well as anti-inflammatory responses. In response to tissue damage, both T-lymphocytes and macrophages start to secrete IL-6 to initiate an immune response. In contrast, the rhythmic release of IL-6 from contracting muscles in response to glycogen depletion induces a systemic anti-inflammatory response [16]. IL-6 is described as a central signal molecule of the acute-phase response [32]. It is the initial cytokine released in the cytokine cascade in response to exercise, and through the stimulation of anti-inflammatory IL-10 and IL-1RA it has inhibitory effects on the pro-inflammatory cytokines TNF-α and IL-1 [26]. According to the characteristics of this cytokine, the increase in IL-6 at NOR state might reflect both the high metabolic demands, as well as the structural tissue damage after training [31,33]. The increase in IL-6 in competitive team sports or after periods of intense training was previously demonstrated. Souglis and colleagues found a prolonged elevation in plasma IL-6 levels lasting up to two days post-match [31,33]. With regard to the use as a biomarker, increased IL-6 levels might reflect the necessity to replenish glycogen stores and, if necessary, restore muscular integrity.

The specific physiological processes, that stimulate the release of IL-17A and IL-6, might be also relevant for the discipline specific release of these cytokines. Although, in badminton players, differences of IL-17A were found between REC and NOR, in soccer players concentrations of IL-6 were different. Although the detailed training contents were not controlled, we assume that the badminton players performed more short-term stop-and-go movements, with a more anaerobic metabolism [34], which cause an inflammatory response. Conversely, the soccer players’ training content indicate a more aerobic metabolism. Comparisons of internal and external load structures of the sport disciplines can prove this in the literature [35,36]. Furthermore, a study investigated the concentration of IL-6 in trained athletes on the arm crank and on the bicycle ergometer after a 90-min interval training. Hoekstra et al. concluded that IL-6 was elevated after both training sessions, but significantly higher on the bicycle ergometer [37]. These results are consistent with those of the present study.

Our results confirm that a habitual loading microcycle affects muscular integrity, indicated by the increase in plasma CK at NOR. CK is a common biomarker also used in sports practice to assess the recovery of muscular performance in athletes of different disciplines [11,19,38]. It is assumed that any type of physical activity is accompanied by a loss of muscular integrity, and the subsequent flooding of intramuscular enzymes into the blood. Specifically, high amounts of accelerations and decelerations, typically found in badminton as well as soccer, are effective here. These movements lead to a high mechanical stress and eccentric force production, which specifically force muscle fiber damage [9,21]. The lack of an increase in LDH may be due to the fact that this enzyme rises quickly after acute bouts of exercise and then falls again, whereas CK is detected in the blood only after a delay and then also over a longer period of time [15].

For CD163 a statistical trend across the disciplines was found (*p* = 0.069). CD163 is expressed on macrophages and monocytes, and may take part in preventing hemoglobin-induced toxicity during physiological and pathological hemolysis [39]. Moreover, CD163 has anti-inflammatory properties, and has been shown to provide protective mechanisms against oxidative stress and myocardial damage [39]. Niemelä and colleagues provide the first evidence that prolonged running increases serum levels of CD163 [39]. They investigated 8 healthy male recreational runners before and after completing a marathon (*n* = 4) or half-marathon (*n* = 4). CD163 increased significantly 3 h after the marathon as well as after the half-marathon. Subjects completing the full marathon distance showed a more pronounced increase in CD163 concentration. After 48 h, a decrease to baseline levels could be observed in 7 out of 8 athletes [39]. It would be interesting to know when the peak of CD163 is reached after load in order to draw conclusions. Despite not being significant, we can confirm the tendency of increased CD163 levels the day after strenuous exercise loads. 

A discipline-specific difference was also found for S100A8. S100A8 is a pro-inflammatory molecule of the S100 protein family and is mainly expressed by myeloid cells [32]. It is known for its role in innate immunity and among other functions organizes cell adhesion and chemotaxis [32]. Together with S100A9 it forms a heterodimer (Calprotectin) and is considered a danger-associated molecular pattern, especially in cardiovascular diseases [22]. Mooren and colleagues conducted an extensive analysis of the exercise response of S100A8/A9 in regard to training status, intensity and type of exercise [32]. A 7-fold increase immediately after the marathon was found, which returned to baseline after 24 h of recovery. We also found a decline the day after the last training session for badminton players. However, these differed significantly from the REC values. It is speculated that these alterations specifically reflect the regulation after a complete training microcycle, and not after an acute exercise response. Other studies found a more or less increase in this inflammatory signaling molecule in response to acute exercise [40,41]. These studies suggest a relation of this cytokine to exercise-induced muscle damage, since eccentric exercise bouts induced a longer-lasting increase in S100A8 [32]. The protein S100A8 is additionally a sensitive marker that is exposed to training-dependent influences [32] and, thus, possible unknown interactions have taken place. In general, the underlying kinetic mechanisms of the training induced S100A8 increase still need to be clarified. 

Additionally, we observed a discipline specific BDNF decrease in plasma concentration at the day after the last training session in soccer players. BDNF is a growth factor, which regulates development, maintenance, and plasticity of neuronal networks [42]. Additionally, BDNF is a key component of the hypothalamic pathway and it is assumed that aerobic exercise has a favorable influence on BDNF-mediated processes [43]. Most studies indicate a transient BDNF increase in response to acute exercise that returns to baseline levels during recovery period of no more than one hour [44,45]. However, our results contradict those of Zoladz et al. [46] and Nofuji et al. [44] Furthermore, the evidence suggests elevation of resting in BDNF concentrations provoked by aerobic exercise [47]. Similar results to ours were found by Wagner et al. after 6-week intense aerobic exercise in healthy young adults [48]. Further, studies have shown lower basal BDNF concentrations in endurance-trained athletes compared to untrained athletes [49]. Thus, it is conceivable that BDNF mediates the described positive effects of regular physical activity on central nerve system structure [50] and function through repeated transient increases in BDNF concentration as a result of each acute soccer training combined with an increase in BDNF utilization capacity. Such a mechanism of action is also known from the IL-6-mediated anti-inflammatory effect of regular physical activity [51]. 

A trend across all athletes was also found for IL-1ß (*p* = 0.06). IL-1ß is secreted as part of a pro-inflammatory immune response and is involved in the induction of acute phase proteins, and in cell proliferation as well as cell differentiation. IL-1ß has been shown to increase not immediately after exercise but after a short delay [52]. In detail, plasma IL-1ß values increased 16-fold after a duathlon race and remained elevated 5-fold at 3 h post exercise cessation [52]. Nonetheless, it may be possible that IL-1ß failed to reach statistical significance due to the presence of IL-10 and IL-1RA, which antagonize the secretion of IL-1ß [52,53].

## 5. Conclusions

The results of the study show that some blood-based biomarkers reflect the recovery and stress status of athletes while many other molecular markers, which have been previously found to be responsive to acute exercise, do not respond to such cycles. CK, IL-6, and IL-17A, showed contrasting regulations across disciplines, while S100A8 seems to be more specific to the badminton players, BDNF may be more suitable for soccer players. Thus, these markers may enable the monitoring of exercise response and recovery cycles in their specific discipline, and subsequently might help coaches and athletes to avoid time lost to overtraining or injury, improve exercise as well as recovery prescription, and to ensure readiness for competition.

Nonetheless, certain limitations have to be considered. One aspect already mentioned is the implementation of microcycles of the training, which were not controlled concerning the detailed training contents. Another aspect is the definition of biomarkers in sports. We only analyzed two time points reflecting ‘extreme values’ on the recovery-fatigue continuum. Future studies should pursue a longitudinal approach with regular biomarker assessment to gain an overall view of individual and seasonal variability [8]. Since we only conducted a group-based analysis, which may be compromised due to responders and non-responders, we are not able to deduce individual reference ranges. Recent research tries to solve this problem with individualized reference ranges using a Bayesian approach and sport specific prior distributions [11]. Furthermore, it should always be considered that recovery-stress responses are multi-faceted phenomena and it may be favorable to utilize a multivariate approach combining physiological and psychological measurements to illuminate recovery and fatigue in a more comprehensive way [1].

## Figures and Tables

**Figure 1 ijerph-18-05776-f001:**
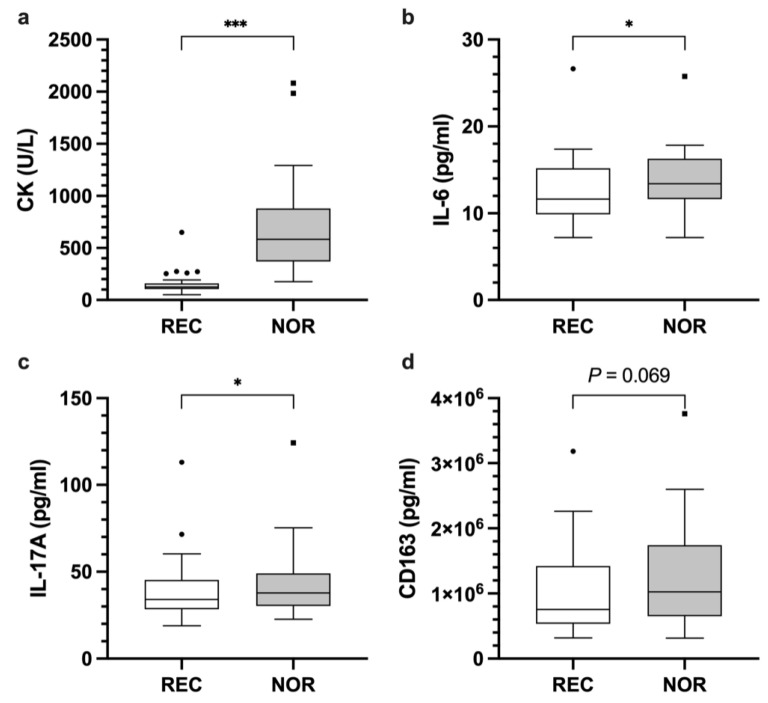
Concentration of CK (**a**), IL-6 (**b**), IL-17A (**c**), and CD163 (**d**) in athletes at REC and NOR time points across disciplines. (*) indicates *p* < 0.05 and (***) indicates *p* < 0.001. Raw data are shown.

**Figure 2 ijerph-18-05776-f002:**
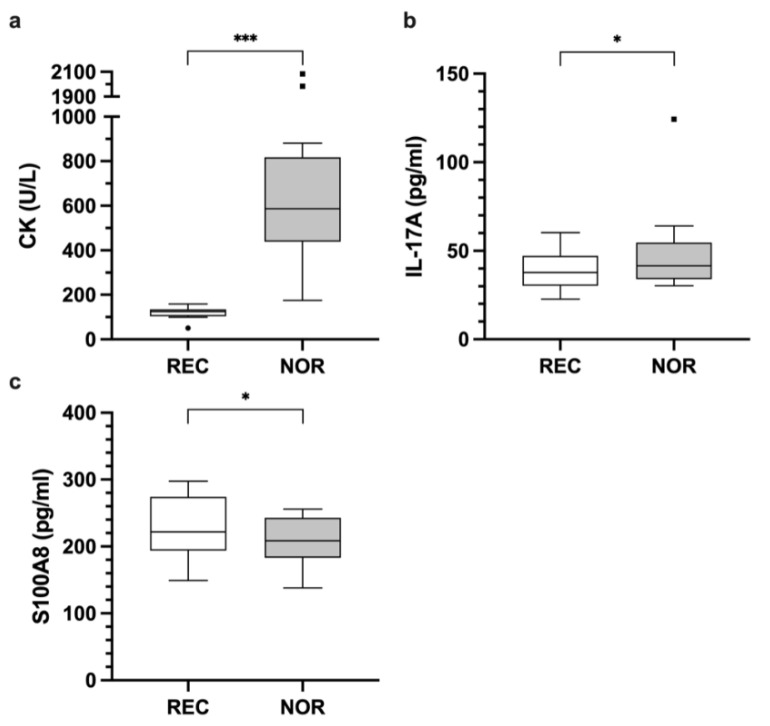
Enzyme activity of CK (**a**), concentration of IL-17A (**b**), and S100A8 (**c**) in badminton players at REC and NOR time points. (*) indicates *p* < 0.05 and (***) indicates *p* < 0.001. Raw data are shown.

**Figure 3 ijerph-18-05776-f003:**
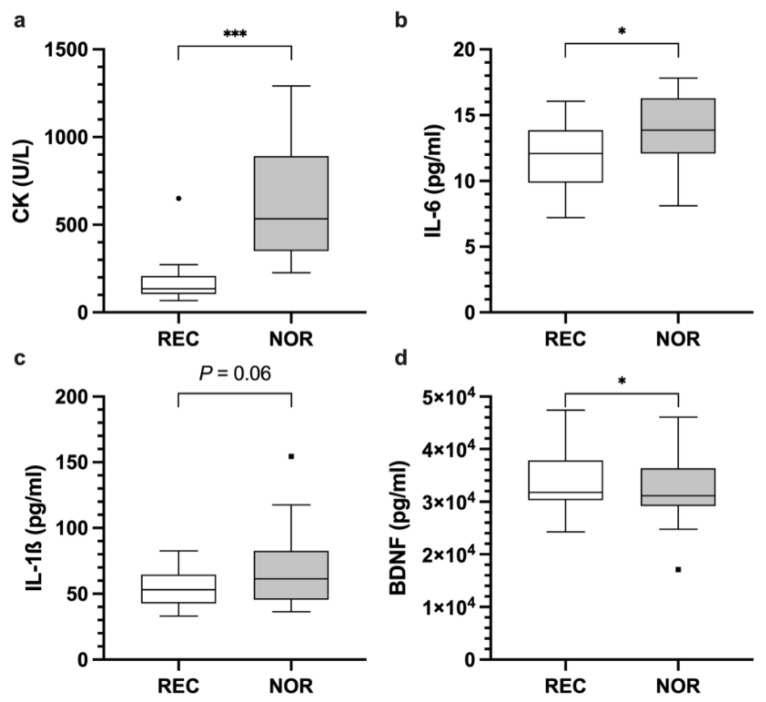
Enzyme activity of CK (**a**), concentration of IL-6 (**b**), IL-1ß (**c**), and BDNF (**d**) in soccer players at REC and NOR time points. (*) indicates *p* < 0.05 and (***) indicates *p* < 0.001. Raw data are shown.

**Table 1 ijerph-18-05776-t001:** Participants’ characteristics.

	Badminton (*n* = 13)	Soccer (*n* = 22)
Age, years	25.6 ± 2.4	28.4 ± 3.4
Height, cm	183 ± 7	180 ± 7
Weight, kg	78 ± 10	79 ± 6
Body fat, %	11.9 ± 3.0	12.4 ± 1.8

Note: Data are presented as mean ± SD.

**Table 2 ijerph-18-05776-t002:** Blood-based biomarker panel and associated categories.

Category	Biomarker
Chemokines	CC-chemokine ligand (CCL)2, CCL4
Cytokines	Interferon-gamma (IFN-γ), interleukin (IL-)10, IL-12p40, IL-17A, IL-1ß, IL-1RA, IL-6, IL-8
Enzymes	Creatine kinase (CK), lactate dehydrogenase (LDH), myeloperoxidase (MPO)
Growth factors	Brain-derived neurotrophic factor (BDNF), glial cell line-derived neurotrophic factor (GDNF)
Hormones	Growth hormone (GH), sex hormone-binding globulin (SHBG)
Other inflammatory signaling molecules	Cluster of differentiation 163 (CD163), S100 calcium-binding protein (S100)A8, S100A12
Proteinases	Matrix metalloproteinases (MMP-)2, MMP-3, MMP-9

**Table 3 ijerph-18-05776-t003:** Concentration of blood-based biomarkers at REC and NOR time points across disciplines (in pg/mL and U/L for CK).

	Recovered		Non-Recovered		Mean Difference, 95% CI	*p* Value
	Mean	SD	Mean	SD		
Chemokines						
CCL2	737.7	229.8	744.3	211.9	6.6 [−47.3, 60.5]	0.804
CCL4	1329.3	163.2	1345.4	157.4	30.1 [−32.3, 77.9] ^W^	0.377
Cytokines						
IFN-γ	173.2	123.6	190.7	151.2	17.4 [−23.5, 58.4]	0.464
IL-10	12.4	5.2	12.9	4.1	0.5 [−0.8, 1.7]	0.444
IL-12p40	2792.8	714.1	2796.5	851.7	18.9 [−371.8, 441.1] ^W^	0.896
IL-17A	38.8	17.7	42.8	19.6	4.7 [0.9, 8.5] ^W^	0.033 *
IL-1ß	56.7	14.3	63.7	25.5	1.7 [−4.5, 11.7] ^W^	0.561
IL-1RA	1130.9	374.7	1196.1	606.5	65.2 [−151.3, 281.7]	0.663
IL-6	12.5	3.8	13.7	3.7	1.3 [0.2, 2.4]	0.024 *
IL-8	42.7	14.5	47.5	17.3	4.7 [−1.7, 11.2]	0.145
Enzymes						
CK	152.6	102.4	693.1	435.5	540.6 [399.2, 681.9]	<0.001 *
LDH	435	387.1	434.6	376.5	−0.4 [−31.4, 30.5]	0.701
MPO	109,757.2	44,962.7	118,082.6	40,440.4	8325.5 [−8074.9, 24,725.8]	0.268
Growth factors						
BDNF	31,282.7	6456.2	30,294.2	6565.3	−988.5 [−2298.5, 321.4]	0.134
GDNF	29.5	8	30.4	9.7	0.9 [−1.8, 3.6]	0.645
Hormones						
GH	2504.4	4223.1	1891	2502	120 [−207, 367.1] ^W^	0.199
SHBG	1.5 × 10^7^	3.2 × 10^6^	1.5 × 10^7^	3.4 × 10^6^	−434,178.6 [−994,313.3, 125,956.1]	0.124
Other inflammatory signaling molecules						
CD163	1.0 × 10^6^	650,929.7	1.2 × 10^6^	794,388.1	174,283.6 [11,188, 337,379.3]	0.069
S100A8	213.4	66.7	204.3	58.7	−5.7 [−27.3, 16]	0.600
S100A12	47,342.8	17,408.7	46,630.2	16,819	−712.6 [−6942.6, 5517.3]	0.818
Proteinases						
MMP-2	292,144.4	37,363.4	297,876	36,001.2	5731.6 [−3788.8, 15,252]	0.230
MMP-3	50,011.2	20,945.4	50,960.5	16,505.7	949.3 [−4812.8, 6711.4]	0.422
MMP-9	114,122.4	84,183.6	115,426.2	89,562.2	1303.8 [−22,467.9, 25,075.6]	0.912

Note: Raw data are shown. SD, standard deviation; ^W^ results of the Wilcoxon signed-rank test; * significant difference between REC and NOR time points (*p* < 0.05).

## Data Availability

The datasets generated during the current study are available from the corresponding author on reasonable request.

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
