# Peer review of "Recovery-Stress Response of Blood-Based Biomarkers"

_ijerph, 2021, doi:10.3390/ijerph18115776_

Round 1
Author Response
Response to Reviewer 1 Comments
This paper presents data on the analysis of selected blood protein biomarkers before and after a training microcycle in n=35 athletes (13 badminton players – blood samples were already obtained in the context of a previous study – and 22 soccer players). The “take-home message“ ist hat CK levels were elevated in response to exercise in both groups of athletes (a well-known piece of knowledge), whereas changes in a few other biomarkers, though reaching statistical significance in some cases, were minor. Although the authors did not do a direct comparison, from the data presented in Fig.3, it looks as if CK baseline levels and degrees of induction were similar in both groups of athletes.
While the paper is well written and the topic is certainly of interest, the design of the study somehow lacks novelty – eventually, it is another “exercise – blood biomarkers – pre/post“ analysis with a small cohort of athletes, even though the authors claim a specific relevance with regard to the “recovery topic“ in the context of training management and control.
Comment 1
The choice of biomarkers somehow appears “random“ – it would probably make more sense to select them from “unbiased“ analyses, such as small-scale proteomics studies with reference groups of athletes and/or a structured meta-analysis of literature data.
Response 1
Reviewer 1 is absolutely right. There is a lack of rationale for the selection of biomarkers in the study. The rationale for the selection of the blood markers was our own studies, which have already shown exercise sensitivity, and which we were able to narrow down by testing the reliability in the context of the exercise cycles. We explained this accordingly in the revision of the manuscript (lines 72-76).
Comment 2
It is unclear why exactly these two discciplines (badminton and soccer) were chosen. In order to gain insight into potential discipline-specific effects (as the authors claim), more disciplines (and larger groups of athletes) should have been analyzed and the respective results directly compared. In addition, while anthropometric characteristics (Tab.1) of athletes from both groups are quite similar, their fitness, i.e. physiological characteristics (VO2max, maximum force etc.) were not assessed (or not presented). Similarly, training microcycles were different for both groups of athletes and not very well standardized, thus, it is unclear if potential difference are intrinsic to the respective groups of athletes or the specific training microcycles in question.
Response 2
The expert's query is justified. We selected the sports according to the fact that they are game sports, one of which contains more endurance aspects (soccer), the other contains shorter fast-paced actions. However, it is true that within the microcycle in the training of competitive athletes, training is not only highly discipline-specific. Therefore, we cannot define the exact training content, but as an advantage we are closer to the training reality. This is certainly a limitation overall, which we have inserted accordingly in the discussion. We have clarified this again in the revised version (lines 211-215, 354-355).
Comment 3
As the authors state themselves, biomarker assessment on the basis of a single “post“ sample, which was obtained at a not very well defined time point in relation to completion of the training microcycle, is not very conclusive, specifically against the background that a lot of the factors the authors analyzed display a tight kinetics in the circulation, with a certain degree of inter and intra- individual variability. Thus, in order to establish them as biomarkers, the analysis of more samples (taken at defined time points after completion of one – or better even, several- training microcycle(s) would be required.
Response 3
Reviewer 1 is completely right. This should only be a first step to narrow down potential biomarkers in the context of microcycles. In the next step, we examine these at multiple time points across seasonal phases. We added this information in the limitations (lines 354-359).
Reviewer 2 Report
General comments
The aim of the study was to analyze the regulation of several physiologic domains including various chemokines, inflammatory cytokines, enzymes, hormones, proteinases and growth factors in athletes recovery-stress training cycles as well as in clarifying their regulation within groups of badminton and soccer players. As mentioned by the authors in their introduction, a problem with the use of biomarker analyses in the context of competitive sport is the lack of reference data and interindividual differences between athletes. For this reason, currently available biomarkers are only partially suitable for training and regeneration management. The study represents a first step towards finding sensitive recovery and stress markers and drawing initial conclusions with regard to two different sport disciplines. In that sense, the study lacks a bit of standardization, certainly in the timing of blood sampling, but it gives reference values for future comparison by others.
Major comments
- While the study presents the advantage to be close to the field, it also presents the disadvantage not to have strictly and quantitatively controlled and monitored the training load and nature/type of the exercises during the microcycles. A questionnaire is not an optimal approach to get the quantification of a training load. So, it is difficult to know what could have caused the differences between badminton and soccer.
- The moment at which blood sampling was performed is not mentioned while it is critical for the interpretation of the data, certainly in the NOR state. How many minutes or hours after the last training session was blood sampling performed? Was this standardized for each athlete? If not, what was the range of duration between the last session and blood sampling between athletes?
Minor comments
- line 100: an outlier was excluded from the analyses because of a too high deviation from the Z score. Please give the magnitude of the deviation. >2 Z-score?
- table 1: give the decimal value for the SD of body fat in the badminton group.
- line 118-110: re-phrase the sentence which is difficult to read.
Author Response
Response to Reviewer 2 Comments
The aim of the study was to analyze the regulation of several physiologic domains including various chemokines, inflammatory cytokines, enzymes, hormones, proteinases and growth factors in athletes recovery-stress training cycles as well as in clarifying their regulation within groups of badminton and soccer players. As mentioned by the authors in their introduction, a problem with the use of biomarker analyses in the context of competitive sport is the lack of reference data and interindividual differences between athletes. For this reason, currently available biomarkers are only partially suitable for training and regeneration management. The study represents a first step towards finding sensitive recovery and stress markers and drawing initial conclusions with regard to two different sport disciplines. In that sense, the study lacks a bit of standardization, certainly in the timing of blood sampling, but it gives reference values for future comparison by others.
Comment 1
While the study presents the advantage to be close to the field, it also presents the disadvantage not to have strictly and quantitatively controlled and monitored the training load and nature/type of the exercises during the microcycles. A questionnaire is not an optimal approach to get the quantification of a training load. So, it is difficult to know what could have caused the differences between badminton and soccer.
Response 1
Reviewer 2 is absolutely right. We took into account that in reality, competitive athletes do not always train in a highly discipline-specific manner. We therefore performed our measurements at defined times during training but did not completely control the content of the training. Therefore, we were very close to reality, but therefore did not directly compare the sports. We have brought this limitation into the discussion again (lines 220-224, lines 354-357).
Comment 2
The moment at which blood sampling was performed is not mentioned while it is critical for the interpretation of the data, certainly in the NOR state. How many minutes or hours after the last training session was blood sampling performed? Was this standardized for each athlete? If not, what was the range of duration between the last session and blood sampling between athletes?
Response 2
This is a valid point. Blood sampling was scheduled on Monday and Friday between 7:45 and 8:30 A.M. REC samples were taken after a minimum of one day of rest. Regarding NOR sampling in the morning of Friday athletes had their last session in the afternoon/evening of Thursday. So, the range of duration between the last session and blood sampling varied between 12 and 16 hours. We added this information to the methods section.
Comment 3
line 100: an outlier was excluded from the analyses because of a too high deviation from the Z score. Please give the magnitude of the deviation. >2 Z-score?
Response 3
The excluded participant yielded a Z-score of 5.75 for IL-6 and 5.74 for S100A8. We added this information to the manuscript.
Comment 4
table 1: give the decimal value for the SD of body fat in the badminton group.
Response 4
Body fat (%) of the badminton group: 11.9 ± 3.0. We added this information to the manuscript.
Comment 5
line 118-110: re-phrase the sentence which is difficult to read.
Response 5
The sentence was re-phrased as follows: Venous blood samples were collected and consisted of 2.7ml as well as 2 x 9ml anticoagulated EDTA plasma vacutainers.
Reviewer 3 Report
The manuscript is well written, but it is recommended to consider the following issues for further improvement.
L24: Could you be more specific on "habitual loading microcycle" for readers?
L53: additional => additionally?
L116: their => the
L142-143: The unit of CK is not concentration and (pg/ml) but enzyme activity and (U/l) as shown in Figure 1a. Please correct them throughout the manuscript including Figures and Tables.
L148: NOR, => NOR;
L151: IL-1R => IL-1RA
L192, 291: regulation => states or other appropriate terms?
L194: other researchers => Please show the references.
L211: regulations => differences or other appropriate terms?
L292: Other studies found a more or less increase of this inflammatory signaling molecule in response to acute exercise. These studies --- => There are no references but please show the evidences such as Eur J Appl Physiol. 2008, 102(4):391-401. doi: 10.1007/s00421-007-0598-1; Int J Sport Nutr Exerc Metab. 2008, 8(3):229-46. doi: 10.1123/ijsnem.18.3.229
Author Response
Response to Reviewer 3 Comments
The manuscript is well written, but it is recommended to consider the following issues for further improvement.
Comment 1
L24: Could you be more specific on "habitual loading microcycle" for readers?
Response 1
The wording means that the micro cycle was not controlled. Thus, the athletes trained freely and the scientists joined them at defined stages of training to collect samples. We have added this in the methods section and as a limitation in the discussion of the revised manuscript.
Comment 2
L53: additional => additionally?
Response 2
Corrected as suggested.
Comment 3
L116: their => the
Response 3
We changed the table description as follows: “Blood-based biomarker panel and associated categories”.
Comment 4
L142-143: The unit of CK is not concentration and (pg/ml) but enzyme activity and (U/l) as shown in Figure 1a. Please correct them throughout the manuscript including Figures and Tables.
Response 4
That is a valid point. We corrected this issue throughout the manuscript.
Comment 5
L148: NOR, => NOR;
Response 5
Corrected as suggested.
Comment 6
L151: IL-1R => IL-1RA
Response 6
Corrected as suggested.
Comment 7
L192, 291: regulation => states or other appropriate terms?
Response 7
We agree with the reviewer and changed the wording from “regulation” to “expression”.
Comment 8
L194: other researchers => Please show the references.
Response 8
We included other research studies to show the differences to our investigation.
Comment 9
L211: regulations => differences or other appropriate terms?
Response 9
We agree with the reviewer and changed the wording from “regulations” to “differences”.
Comment 10
L292: Other studies found a more or less increase of this inflammatory signaling molecule in response to acute exercise. These studies --- => There are no references but please show the evidences such as Eur J Appl Physiol. 2008, 102(4):391-401. doi: 10.1007/s00421-007-0598-1; Int J Sport Nutr Exerc Metab. 2008, 8(3):229-46. doi: 10.1123/ijsnem.18.3.229
Response 10
We agree with the reviewer and included both suggested studies.
Round 2
Reviewer 1 Report
The manuscript has been improved by adding more information and evaluating limitations in a more critical manner.